# Using a Socio-Environmental Approach to Explore the Determinants for Meeting the Recommended Physical Activity among Adults at Risk of Diabetes in Rural Indonesia

**DOI:** 10.3390/healthcare9111467

**Published:** 2021-10-29

**Authors:** Fransiskus Xaverius Widiantoro, Jing-Jy Wang, Yi-Ching Yang, Cheng-Chen Chou, Chi-Jane Wang

**Affiliations:** 1Department of Nursing, School of Health Sciences Saint Borromeus, Bandung 40122, Indonesia; widiantoro@stikesborromeus.ac.id; 2Department of Nursing, College of Medicine, National Cheng Kung University, Tainan 70101, Taiwan; ns127@mail.ncku.edu.tw; 3Department of Medicine, College of Medicine, National Cheng Kung University, Tainan 70101, Taiwan; yiching@mail.ncku.edu.tw; 4Institution of Community Health Care, College of Nursing, National Yang Ming Chiao Tung University, Taipei 11221, Taiwan; ccchou@nycu.edu.tw; 5Nursing Department, National Cheng Kung University Hospital, Tainan 70403, Taiwan

**Keywords:** prediabetic state, physical activity, social environments, culture

## Abstract

Moderate-to-vigorous physical activity (PA) is recommended to mitigate the risk of diabetes. This study explored the PA of adults at risk for diabetes in rural Indonesia and determined the requirements for meeting the recommended PA level. In total, 842 adults were screened using a diabetes risk test in a rural health centre; among them, 342 were at risk of diabetes. The level of PA was assessed using the International Physical Activity Questionnaire, whereas the associated factors underlying the three domains –individual, support, and environment—were determined by the Influences on Physical Activity Instrument. The data analysis included a three-step multiple linear regression (MLR) and logistic regression (LR). Overall, 40.6% of the participants met the recommended PA. According to the MLR analysis, among males, individuals who gave PA a higher priority and had enough time to perform PA were predicted to have a higher activity energy expenditure (MET-minutes per week). According to the LR analysis, men were more likely to meet the recommended PA, and people who gave PA a lower priority and had less access to space for PA were less likely to meet the recommended PA level. Strategies for promoting PA in rural Indonesia include focusing on women, people who prioritize PA less, and those who have less time and space in which to be physically active.

## 1. Introduction

Nearly 1 in 10 adults worldwide will suffer from Type 2 diabetes mellitus (T2DM) by 2045, with one-third living in rural areas in less developed countries, such as Indonesia [1]. Diabetes has been a major health problem in Indonesia since the early 1980s [2]. With over 10 million adults with diabetes, Indonesia ranked seventh in the world in 2017 [1]. According to these estimates, the prevalence of diabetes is expected to continue in Indonesia [3]. To address the alarming increase in the number of Indonesian adults with diabetes, the prevention of diabetes is crucial.

Not meeting the recommended level of physical activity is a leading risk factor for T2DM, with nearly one-quarter of adults worldwide identified as being inactive [4]. Increasing physical activity is an important strategy in diabetes prevention [5] and an often-targeted intervention because it is modifiable [6]. The age-adjusted prevalence of insufficient physical activity is 22.6% in Indonesia [7], with an average of 3513 steps per day, which is lower than the global average of 5000 steps per day [8]. Diabetes and insufficient physical activity are major health problems in Indonesia. To date, the prevention of diabetes in Indonesia has been explored for urban areas [9], but there is a dearth of research related to rural areas.

The influence of physical activity on diabetes risk needs to be determined so that interventions can be appropriately targeted. It is well known that meeting the recommended level of physical activity dramatically lowers the risk of death [10]. However, initiating and maintaining regular physical activity for people at risk of diabetes mellitus is difficult [11]. Reasons for not adhering to the guidelines include lack of self-motivation [12], lack of time [13], lack of self-monitoring or feedback [14], lack of necessary resources and the presence of environmental barriers [15], and socioeconomic limitations [16]. There is a litany of determinants and antecedents of physical activity. Thus, it should not be explored simply as a problem at the individual level; rather, a societal perspective is needed to address how to increase physical activity. The ability to engage in physical activity also depends on policies, the environment, and cultural norms. Although these factors have been widely studied in many developed countries [12,13,14,15,16], they have not been systematically investigated in less developed countries, particularly in Indonesia, and for adults at risk of diabetes living in rural areas with limited healthcare access.

The Indonesian population consists of many ethnic groups, each having unique customs and cultural practices. Indonesia is a culturally diverse society, with over 87% of people identifying as Muslims on the 2019 census [17]. Gender plays an especially important role in societal expectations [18]. Regarding the impact of religion on physical activity, a survey conducted across 38 countries found that the prevalence rate of not meeting the recommended level of physical activity in the Muslim world is higher than in non-Muslim countries [19]. Furthermore, the prevalence of inactivity among adult females (35.5%) is 1.4 times greater than that of adult males [20]. Men participate in sports and exercise more than women do, primarily due to sex/gender roles [21]. Because each country has its own social and cultural influences, different genders bear different responsibilities. Among the sociocultural factors, family obligations are the strongest barrier to engaging in regular physical activity. Women frequently take on the role of assuming most of the family’s obligations. The religious beliefs and values of Muslim women impact the manner in which they manage their lives. Their religion expects them to wear modest dresses and to remain segregated from men [22]. It can be concluded from the research that the limitation of physical activity for Muslim women can be attributed to cultural norms, including gender roles and religious beliefs. A ‘unique social-cultural living environment’ perspective should be included when exploring the influencing factors of physical activity for underserved populations in developing countries.

Bauman and colleagues [23] developed a social-ecological model, which appraised the evidence to confirm the factors affecting physical activity. The approach not only addresses individual characteristics but also considers the social and physical environment context, which includes family, friends, and the design of the community environment and facilities [24]. The model offers an overarching framework for comprehending the influences on physical activity to identify physical activity behaviours of the community, rather than having a traditional, isolated focus on intrapersonal factors [25]. Thus, the Influences on Physical Activity Instrument (IPAI), a multi-level instrument that contains three domains (individual, support, and environment) [26], was applied to the present study to identify the determinants of physical activity for a distinct adult population of Muslims residing in rural Indonesia, who are at risk of diabetes.

The study focused on two research questions: (1) Which influencing factors can predict whether rural adults significantly increase their weekly activity energy expenditure (in MET-minutes)?; (2) Which factors are significantly related to rural adults meeting the recommended physical activity level? 

The findings may assist health professionals in defining appropriate strategies, as well as the Indonesian government in making policies that encourage physical activity among adults, especially populations at risk of diabetes, living in rural areas, and restricted by certain cultural norms.

## 2. Material and Methods

The study took place in an area that was served by a public health centre in West Java, Indonesia. The area has an approximate population of 12,000, the majority of whom are Muslim. This public health centre possesses limited resources but offers some primary healthcare services, including physical examination and chronic illness management. Most of the residents in this rural area have low education and income levels.

### 2.1. Participants

The required sample size was calculated as 340–355, assuming a 26% prevalence of physical inactivity [27] with a 5% relative precision, 95% confidence interval (CI), and a 15–20% attrition rate. A community-based sample of adults at risk of diabetes was referred by the public health centre in rural West Java.

For the diabetes risk test, the current study used systematic random sampling to select 842 families from the community registration household list, and one adult was randomly selected from each family to perform the diabetes risk screening test. The 842 adults were screened by the public health physician using a seven-item risk test by the American Diabetes Association [28], and 342 scored higher than 4. These individuals were invited to participate in the study. The exclusion criteria were: a diagnosis of diabetes mellitus, communication impairments, and the inability to move freely due to physical deformity.

### 2.2. Measures

This study used three measurement tools to collect the data: a sociodemographic and economic information self-report, the International Physical Activity Questionnaire Short-Form (IPAQ-SF), and the Influence on Physical Activity Instrument (IPAI).

The self-reported socio-demographic and economic data were used to assess the participants’ age, gender (male/female), educational level (elementary or below/junior high or above), marital status (single/married), employment (unemployed/employed), and family history of T2DM (diagnosed by a physician). Employment was defined as working ≥ 40 h per week per Indonesian regulations.

#### 2.2.1. IPAQ-SF for Measuring Physical Activity Levels

The English version of IPAQ-SF is a seven-item self-report of time spent on and frequency of moderate and vigorous activities in job-related, travel, family, and recreational pursuits in the previous 7 days [29]. Activity is categorized as low, moderate, or high. The IPAQ-SF is used in 12 countries.

Sutoyo et al. [30] developed an Indonesian version of the IPAQ-SF that is valid for measuring the physical activity of adolescents with obesity. In the current study, a convenient sample of 78 Indonesian adults who were overweight or obese was recruited to examine the internal consistency of the IPAQ-SF-Indonesian version. Cronbach’s alpha ranged from 0.77 to 0.91.

Regarding the recommended level of PA, the current study defined the participants who reported at least 150 min of moderate or 75 min of vigorous activity per week as the ‘active’ group, meeting the activity level recommended by Healthy People 2020 [31]. Conversely, those who fell below the criteria were defined as the ‘inactive’ group.

#### 2.2.2. IPAI to Explore the Influences on Physical Activity

The IPAI was used to explore the predictors of physical activity in the current study. Donahue et al. [26] developed the IPAI to study people at risk for diabetes to identify the barriers and supports for meeting a recommended level of physical activity. The IPAI is a multi-dimensional instrument. The original version feature 21 items that assess three domains (individual, support, and environment), with six subscales/factors containing two to five items each. The individual domain contains three subscales: low priority (five items), low weight control benefits (two items), and injury concerns (four items). The support domain contains one five-item subscale (support). The environment domain includes two subscales: place for activity (three items) and time for activity (two items).

The items are rated on a 4-point Likert scale, ranging from 1 (strongly agree) to 4 (strongly disagree). To compute the score of each IPAI subscale, all negatively worded item scores were reversed and added. Lower scores on IPAI subscales indicated fewer barriers to engaging in physical activity in that dimension.

The construct validity of the IPAI was examined by Donahue et al. [26] by using exploratory factor analysis, which resulted in keeping 18 out of the 21 items. The reliability (internal consistency) of the six subscales was confirmed by Cronbach’s α ranging from 0.53 to 0.77.

A culturally appropriate Indonesian version of the IPAI (IPAI-I) was created through translation validation and construct validity testing [32]. The three-domain model of the 21 item IPAI-I features acceptable construct validity, as verified through exploratory and confirmatory factor analysis using 682 Indonesian adults. The reliability for each domain was examined by test-retest (0.73–0.92) and internal consistency (Cronbach’s α = 0.82 to 0.91).

#### 2.2.3. Procedures

After obtaining approval from the Ethics Review Board of the local government, each participant was given a written informed consent letter before completing the questionnaires in the public health centre. In the process of collecting the demographic data and administering the IPAQ and IPAI-I in the public health centre, due to the high illiteracy rate among the participants, three trained interviewers read each question to the participants and recorded their answers. There were no missing data in the questionnaires. The interviewers’ inter-rater reliability prior to data collection was satisfactory, at κ = 0.88.

#### 2.2.4. Data Analysis

SPSS for Windows version 19.0 (Armonk, NY, USA: IBM Corp.) was used to analyze the data. The rate of not meeting the recommended level of physical activity was calculated for the 342 participants at risk of diabetes. Before conducting the linear regression analysis, all the continuous variables were tested for normality, in which weekly MET-minutes were converted to z-scores for accurate estimation. Multiple linear regression analyses were performed simultaneously to assess the relationships between the demographic characteristics and the influences on physical activity and the energy expenditure of physical activity (weekly MET-minute). In all the analyses, age and influences on physical activity (priority, weight control, no injury concern, social support, time, and place) were treated as continuous measures; all the other factors were categorical variables. The first regression analysis included only demographic characteristics. The second analysis included the influences on physical activity. The third model (full model) included all the demographic characteristics and influences on physical activity.

Logistic regression was used to examine how the demographics and influences on physical activity were associated with meeting versus not meeting the recommended physical activity level. The rate of odds ratios (RR) and confidence intervals (CI) were reported to show statistical significance (*p* < 0.05).

## 3. Results

### 3.1. Demographics, Health Factors, IPAI, and PA Levels

The mean age of the 342 participants was 35.8 (standard deviation = 10.7). Almost all of the participants were Muslim (*n* = 315, 92.1%). A majority of them were female (*n* = 199, 58.2%) and aged 20–34 (*n* = 214, 62.5%). Most were married, educated to a junior high level, and unemployed (*n* = 199, 58.2%). Women reported a higher unemployment rate (*n* = 164/199, 82.4%) than men (*n* = 35/143, 17.6%). Among the unemployed, women tended to be homemakers, and men were mostly retired or had quit their jobs temporarily. Most of the men were employed as office workers or worked in factories or farms. Over 80% of the participants reported a family history of diabetes and hypertension. Two-thirds (66.7%) were overweight or obese (Table 1).

In the participants’ self-reported data for each domain in the IPAI, the item mean scores (2.31–2.57) and modes (2.0–2.57) indicated a substantial barrier to engaging in physical activity. In terms of physical activity patterns, 56.7% of the 342 participants achieved a minimum of 600 MET-min/week and their physical activity was classified as ‘sufficient’ by the IPAQ. In addition, applying the definition of recommended physical activity level, 40.6% (95% CI = 39.8–41.4) of the participants were assigned to the ‘physically active’ group. However, the average daily sitting time for all the participants was 2.7 h (range, 1–6; 95% CI = 2.6–2.8) (Table 2).

### 3.2. Bivariate Analysis for the Factors of Meeting Recommended PA

In the bivariate analysis, men (51.0%) were more active than women (33.2%), with a higher rate of odds ratio for meeting the ‘physically active’ level (RR= 2.1; *p* = 0.001). There was no difference between the active and inactive groups for any other demographic variables or health factors. However, lower prioritization of physical activity (RR= 0.4; *p* < 0.001) and lack of place for physical activity (RR = 0.55; *p* < 0.05) were associated with a lower rate of odds ratio for meeting the recommended level of physical activity (Table 3).

Men were more likely to engage in high-intensity levels of physical activity and to expend more total physical energy than women. The difference in physical activity levels between males and females was further analyzed to determine the attributes of the level of intensity (vigorous, moderate, and walking) (Figure 1).

### 3.3. Logistic Regression to Determine the Factors for Meeting the ‘Physically Active’ Level

A logistic regression analysis using the forward method indicated that men were more likely to engage in physical activity than women (adjusted RR= 2.11; 95% CI = 1.34–3.31; *p* = 0.002). The participants who gave physical activity a lower priority (adjusted RR= 0.44; 95% CI = 0.24–0.81; *p* = 0.013) and had less access to places for physical activity (adjusted RR= 0.55; 95% CI = 0.32–0.95; *p* = 0.045) were less likely to be active. The three variables accurately predicted 63.2% of the physical activity behaviours of the 342 participants at risk of diabetes (Table 3).

### 3.4. Multiple Linear Regression to Determine the Predictors of Expendituring Higher Energy

The results of the multiple regression models after transforming MET-min/week into z-scores are shown in Table 4, along with the standardized beta coefficients and significance levels. In Models 1 and 2, gender (male vs. female) had a positive effect on weekly MET-minutes through physical activity (β = 0.281, *p* < 0.01), while individuals who gave a lower priority to physical activity had a significantly negative level of activity energy expenditure (β = −0.15, *p* = 0.012). The combined model showed that gender had a significantly positive effect on weekly MET-minutes through physical activity (β = 0.193, *p* < 0.01), while lower priority and lack of time for physical activity had a negative effect on energy expenditure (β = −0.122, *p* = 0.025; and β = −0.127, *p* = 0.034, respectively). The full model explained 10.7% of the variance.

## 4. Discussion

We examined the influences on physical activity among adults at risk of diabetes in a rural area of West Java of Indonesia, where a majority of the residents are Muslim. The results show that less than half (40.6%) of the participants reported meeting the Healthy People 2020 objectives for physical activity. The participants differed in several potentially modifiable characteristics as functions of the reported activity levels. Those who were male, who gave physical activity a high priority, and who had access to a place for physical activity tended to be more active. Furthermore, those who made time for physical activity expended much more physical energy.

In the current study, more than half of the participants (59.4%) did not meet the recommended level of physical activity. This percentage is lower than that of diabetes patients in general [9,33] but higher than the rate of the general Muslim population [19], Indonesian adults [27], and the worldwide adult population. Physical inactivity is more prevalent in Muslim countries than in non-Muslim countries [19]. Clearly, religion and socio-cultural factors have a potential impact on physical activity performance, and such an impact needs to be considered in the development of appropriate strategies for community health programs for adults at risk of diabetes when promoting high-intensity levels of physical activity with this group.

The social-ecological model considers an individual’s behaviour as being influenced by multiple variables in a larger context and emphasizes the effects of the environment on physical activity. Thus, environmental changes should be prioritized to promote more active lifestyles [34]. In the current study, the IPAI was used to assess the influences on physical activity. The instrument contains six multi-item scales that explore the individual, support, and environment domains. The results show that individual and social environmental factors, including personal priority, place, and time, contribute to physical activity performance. Similar findings have also been reported, in which a person’s priority for physical activity is treated as a form of autonomous motivation [12,26]. Barriers to physical activity often include attitudes and beliefs about the activity, and autonomous motivation is a significant determinant of regular participation in the activity [35]. Physical education [36] and peer support [37] can increase autonomous motivation to encourage individuals to prioritize physical activity.

In terms of the environmental effects on physical activity, the present study’s results demonstrate that having access to space for physical activity can offer people a better chance of reaching the recommended activity levels, and making time for physical activity allows people to expend more total physical energy. Most rural areas feature low residential density, low street connectivity, and a lack of transportation options and lighting [38], but they tend to have more available space to exercise compared with urban settings. It is a challenge for community professionals to create an accessible space and to make time to increase the physical activity levels of adults at risk of diabetes who reside in rural areas with limited resources. Kahan [19] designed a cultural program to significantly promote physical activity among Somali women. Cultural activities and places of worship should be considered as intervention program strategies.

The evidence suggests that among older adults, those with more social support for physical activity are more likely to engage in leisure physical activity, especially when social support comes from family members [39]. However, the social support variable (resources including family and friends) dropped out of the model for this population after controlling for other variables. In other words, social support had no effect on physical activity among adults at risk of diabetes in Indonesian rural areas. The Indonesian culture emphasizes that family is one’s social responsibility and that the younger generation has a duty to take care of the older generation in the family. To provide financial assistance, young people may be away from home for work or sacrifice their leisure time to support their siblings. This pressure can restrict the potential growth of young Indonesians, as much of their time is spent on family duties. The positive impacts of family satisfaction, family interaction, and family stability come from families participating in leisure activities together [40]. It is a challenge to design a program that allows Indonesian family members to get together and support each other in leisure activities, especially for lower-social-class families in rural areas.

Regarding the social demographic effects on physical activity, gender demonstrated a significant effect in the current study. We found that men were more likely to engage in physical activity and expended more total physical energy than women. Additional analysis was performed to determine how men and women differed in the intensity of their physical activity (vigorous, moderate, and walking). In less developed countries, women usually play multiple roles and take on many family obligations, while men go out to work because of socio-cultural expectations [8,20,41], although women account for a large proportion of the workforce [18]. However, in western countries, women in rural areas who are Caucasians and have higher educational achievements tend to demonstrate higher intensity levels of physical activity [42,43]. There is no doubt that gender equality has a significant effect on increasing leisure-time physical activity among women [44].

In addition to cultural effects, social role expectations have a negative effect on women’s physical activity levels. Some of these expectations come from religious norms that form a barrier for females, particularly Muslim women, making it difficult for them to participate in sports [45,46]. Specific barriers include modest clothing that is not suitable for physical activity, religious rules that require women to be chaperoned in public, and the lack of segregated fitness facilities for women [47,48]. This phenomenon is particularly common in rural areas. These barriers are found in Islamic countries such as Indonesia and in Muslim-minority countries [49].

Thus, if healthcare professionals aspire to design effective physical activity programs for lower-social-class women, they should keep in mind that housework is time-consuming [50]. Useful strategies for promoting PA in rural communities include a multilevel approach to address priorities in life [51] or designing cultural/religious events and practices to include physical activities [52], which may help to promote high-intensity physical activity for the population segments that are at risk of inactivity, particularly women and people in lower socioeconomic groups.

Other than gender, no variables in this study showed a significant difference between the activity groups, including socio-demographic characteristics and health histories (e.g., hypertension or obesity). This finding contradicts the statistically significant positive associations that other studies have found [42,53,54]. As these potential markers for activity are also risk factors for pre-diabetes, the lack of significant differences was most likely due to the homogeneity of the pre-diabetes population.

The limitations of this study include the following. As the data were collected by self-report and the design was cross-sectional, the evidence is limited and cannot be used to ascertain whether the difference between the active and inactive groups indicates a causal relationship. Furthermore, individuals may overstate their physical activity intensity levels. The findings of this study only serve as a reference for adults living in rural areas of less developed countries, especially where it is traditional for women to take care of most of the household chores, especially Muslim women.

## 5. Conclusions

For areas with a variety of cultural impacts in Indonesia, PA promotion strategies should first focus on women, people who give PA a lower priority, and those who have less time and space for being physically active. Research must consider the implications of these findings for people whose age is similar to those of the study’s participants, who were in the most productive years of their lives, as well as people with similar sociocultural backgrounds. Additionally, when applying a socio-ecological approach to similar populations in developing countries to increase physical activity, the approach needs to be adjusted to better fit the population’s characteristics. Gender differences should be concerned when designing exercise promotion projects. It would be valuable to apply qualitative methods to further explore gender issues in related fields in future research.

## Figures and Tables

**Figure 1 healthcare-09-01467-f001:**
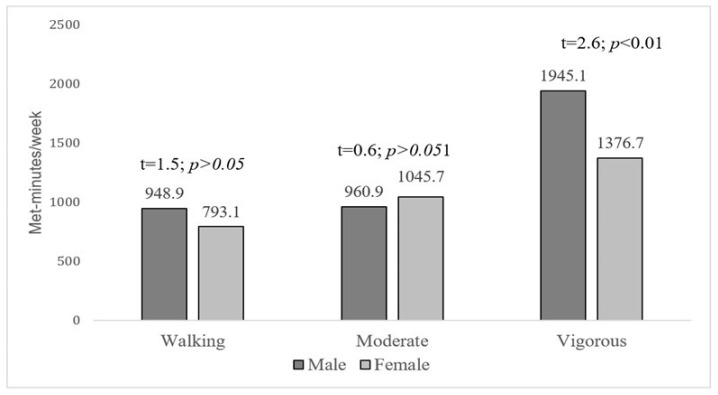
Comparison of energy expenditure between men and women in each intensity level of physical activity.

**Table 1 healthcare-09-01467-t001:** Distribution of demographic characteristics and health factors (*n* = 342).

Variable	Group	Distribution
Demographic Factors		Mean (SD)
Age (y/o)		35.8 (10.7)
		**N (%)**
Gender		
	Male	143 (41.8)
	Female	199 (58.2)
Education		
	Elementary or under	154 (45.0)
	Junior high or above	188 (55.0)
Employment		
	Unemployed	155 (45.9)
	Farmer	75 (21.9)
	Employee	107 (32.2)
Marital status		
	Single/divorced/widowed	49 (14.3)
	Married	293 (85.7)
**Health Factors**	**Group**	**N (%)**
Family history of DM		
	Yes	284 (83.1)
	No	58 (16.9)
Hypertension		
	Yes	297 (86.8)
	No	45 (13.2)
Overweight or obese		
	Yes	228 (66.7)
	No	114 (33.3)

**Table 2 healthcare-09-01467-t002:** Distribution of influences of physical activity and physical activity behaviours (*n* = 342).

Variable		Distribution	
Influence of Physical Activity (IPAI)	Full Scale	Item Mean (SD)	Mode
**Individuals**			
Low priority	1–4	2.34 (2.20)	2.0
Low weight control benefit	1–4	2.57 (2.50)	2.5
Injury concerns	1–4	2.31 (2.25)	2.0
**Support**			
Little support	1–4	2.49 (2.40)	3.0
**Environment**			
Fewer places for activity	1–4	2.39 (2.33)	2.67
Less time for activity	1–4	2.50 (2.50)	2.57
**Physical activity**	**Group**	**N (%)**	
MET-min/week ≥ 600			
	Yes	194 (56.7)	
	No	148 (43.3)	
Recommended levels ^a^			
	Yes	139 (40.6)	
	No	203 (59.4)	
MET-min/week		Mean (SD)	
Total ^b^		1448.7 (1599.0)	
Vigorous intensity (*n* = 128)		1691.9 (1252.6)	
Moderate intensity (*n* = 131)		1002.9 (786.4)	
Walking (*n* = 172)		846.7 (637.6)	
Daily duration (hour) for sitting (*n* = 342)		2.7 (1.4)	

^a^ Recommended level is met with at least 150 min of moderate or 75 min of vigorous activity per week. ^b^ Total: MET-min/week is calculated by summing mild, moderate, and vigorous activities.

**Table 3 healthcare-09-01467-t003:** The rate of ‘physically active’ levels, the rate of odds ratios (RR) from the bivariate analysis, and the adjusted RR (ARR) from the logistic regression were used to determine the factors for meeting the recommended physical activity level (*n* = 342).

Variable	Rate of ‘Physically Active’ Level (%)	RR	B (95 % CI of RR) ^a^	ARR	B (95 % CI of ARR)
**Demographic Factor**					
Age [y/o, M (SD)]	36.3 (11.2)	0.99	0.01 (0.97–1.01)	0.99	0.01 (0.98–1.01)
Gender					
Female	33.2	1			
Male	51.0	2.10 **	0.74 (1.35–3.27)	1.79 *	0.58 (1.01–3.18)
Education					
Elementary or under	36.4	1			
Junior high and above	44.1	1.38	0.32 (0.89–2.14)	1.19	0.19 (0.62–2.29)
Employments					
Unemployed	31.9	1			
Farmer	48.0	2.04 *	0.71 (1.16–3.58)	1.84	0.63 (0.93–3.64)
Employee	48.6	2.13 **	0.75 (1.28–3.52)	1.27	0.23 (0.62–2.61)
Marital status					
Single	32.7	1			
Married	42.0	1.49	0.42 (0.79–2.83)	1.71	0.55 (0.86–3.40)
**Health factor**					
Family history of DM					
Yes	57.4	1			
No	58.6	1.04	−0.04 (0.58–1.84)	1.12	−0.10 (0.59–2.15)
Hypertension					
Yes	41.1	1			
No	37.8	0.87	−0.14 (0.46–1.66)	0.75	−0.22 (0.37–1.52)
Obese					
Yes	39.5	1			
No	46.4	1.33	−0.18 (0.75–2.36)	1.07	−0.26 (0.55–2.10)
**IPAI factors**	**Mean**				
**Individual**					
Low priority	2.27	0.40 **	−0.91 (0.22–0.74)	0.46 *	−0.85 (0.23–0.92)
Low weight control benefits	2.58	0.82	−0.20 (0.48–1.39)	0.89	−0.11 (0.48–1.61)
Injury concerns	2.26	0.76	−0.28 (0.45–1.27)	0.96	−0.8 (0.52–1.77)
**Support**					
Little support	2.44	0.63	−0.46 (0.39–1.01)	0.83	−0.2 (0.46–1.52)
**Environment**					
Few places for activity	2.45	0.55 *	−0.60 (0.33–0.92)	0.55 *	−0.39 (0.32–0.95)
Less time for activity	2.33	0.69	−0.37 (0.43–1.10)	0.84	−0.18 (0.49–1.45)

^a^ 95% CI: confidence interval of 95%; * *p* < 0.05; ** *p* < 0.01.

**Table 4 healthcare-09-01467-t004:** Multiple linear regression results for the factors of total MET-min/week using z-scores (*n* = 342).

Variable	MET-Min/Week (Standardized B)		
	Model 1	Model 2	Model 3
**Demographics factor**			
Age	−0.095		−0.100
Gender (male/female)	0.181 **		0.193 **
Marital status	0.058		0.066
Education	0.066		0.019
Working status			
Farmer/Uemployed	0.059		0.086
Employee/Unemployed	0.043		0.038
**IPAI factor**			
**Individual**			
Low priority		−0.150 *	−0.122 *
Low weight benefits		−0.079	−0.082
Injury concerns		−0.019	−0.017
**Support**			
Little support		−0.033	−0.013
**Environment**			
Fewer places for activity		−0.023	−0.020
Less time for activity		−0.109	−0.127 *
Adjusted R^2^	7.0%	4.7%	11.0%

* *p* < 0.05; ** *p* < 0.01.

## Data Availability

The data presented in this study are available on request from the corresponding author. The data are not publicly available for ethical reasons.

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
