# Peer review of "Using a Socio-Environmental Approach to Explore the Determinants for Meeting the Recommended Physical Activity among Adults at Risk of Diabetes in Rural Indonesia"

_healthcare, 2021, doi:10.3390/healthcare9111467_

Round 1
Reviewer 1 Report
This is an interesting area to examine the factors that might predict physical activity levels amongst the Indonesian population. My main concern is the representativeness of the population and the authors need to make the case for this. Additionally the IPAQ parameters are not normally distributed and will require transformation or non-parametric analysis.
On a minor note the authors conclude that religion is a major factor, however this is rather simplistic and it is better to talk about the impact of sociocultural factors rather than a religion.
Also the beta coefficients should be given with the respective 95% CI (Table 3).
Author Response
Dear Reviewer 1:
Thank you for your comments and suggestions. I have based on your comments to revise the relevant context. In addition, we invited an English Professional to edit the manuscript. For your comments, we have answered these questions in the table below. Looking forward to hearing from you ASAP.
Thank you for giving us the opportunity to do the major revision.
Kindly Regards
Corresponding author:Chi-Jane Wang, NCKU, Taiwan.
Table. Responding to reviewer's comments
|
No |
Reviewer Comments |
Authors’ Response |
|
1. |
This is an interesting area to examine the factors that might predict physical activity levels amongst the Indonesian population. |
Thank you for your approval and encouragement. |
|
2. |
My main concern is the representativeness of the population and the authors need to make the case for this. |
Thank you for pointing this out. This is a very important issue for prevalence research. In the current study, 342 participants at risk of diabetes were screened from the 842 rural adult residents through systematic random sampling. This is a community-based study, so it can only represent the area where we sampled (a rural area in West Java, Indonesia), where the majority of the residents are Muslim. The reasons for sampling the population have been illustrated in the Introduction, page 3, lines 18-21. Furthermore, the application of the findings is mentioned in the Discussion, page 8, line 5-6, and page 10, line 31-33 where we described the study’s limitations. |
|
3. |
IPAQ parameters are not normally distributed and will require transformation or non-parametric analysis. |
Thank you for your suggestion. Before running the linear regression in this study, the continuous data including the MET-min/week had been tested for a normalized distribution. The dependent variable is closed to normalized distribution. To improve the accuracy estimation, MET-min/week were also converted into z-scores in the linear regression analysis according to your suggestion. However, compared with the results of the unconverted method, the significant predictors did not change, but there was a small difference between the new parameters. The content has been revised in the Methods section and Results section. Please see page 4, lines 37-39 and page 7, Section 3.4, and Table 4.. |
|
4. |
On a minor note the authors conclude that religion is a major factor, however this is rather simplistic and it is better to talk about the impact of sociocultural factors rather than a religion. |
Thank you for your suggestion. According to your suggestion, the term was revised with sociocultural factors. Please see page 10, lines 40. |
|
5 |
the beta coefficients should be given with the respective 95% CI (Table 3). |
Thank you for pointing this out. The 95% CIs for the beta coefficients have been added to Table 3. |

Reviewer 2 Report
The aims of the study are clearly defined and the study design and methods are adequated to achieve those.
There are a few issues that deserve attention from the authors:
Introduction
Page 2, third paragraph:
From a cultural perspective, family obligations are the strongest barrier to engaging in regular physical activity. Regarding the different effects of sport and exercise on men and women, it has long been established that the differences come from the stereotypes on sex/gender roles [21].
Do you mean that effects of PA on men and women depend on gender roles, that engagement in PA depend on gender roles or that perception of PA depend on gender roles? Please clarify.
Please check the last paragraph on the same page:
The current study had two research questions: 1) What are the factors that can be predicted who expended more activity energy in terms of MET-minutes per week among adults in rural areas?;
Methods
Page 3, Measures,
2.2.2. Influence of Physical activity. Please check.
Results, page 5, second paragrapgh
In terms of physical activity patterns, 56.7% of the 342 participants achieved a minimum of 600 MET-min/week and were classified as ‘sufficiently active’. However, 59.4% (95% CI = 56.43–62.37) of the participants were not considered physically active.
Please check.
In the bivariate analysis, men (51.0%) were more active, with a higher prevalence ratio (PR = 2.1; p = 0.001) than women (33.2%).
Higher prevalence ratio of meeting PA recommendations?
Discusion, page 9, fourth paragraph,
In other words, social support had no effect on physical activity among adults with diabetes in Indonesian rural areas.
Did you explore the model in men and in women?
Some additional qualitative research on this issue would be welcome.
Author Response
Dear Reviewer 2:
Thank you for your comments and suggestions. I have based on your comments to revise the relevant context. In addition, we invited an English Professional to edit the manuscript. For your comments, we have answered these questions in the table below. Looking forward to hearing from you ASAP.
Thank you for giving us the opportunity to do the major revision.
Kindly Regards
Corresponding author:Chi-Jane Wang, NCKU, Taiwan.
Table. Responding to reviewer's comments
|
No |
Reviewer Comments |
Author Responses |
|
1. |
The aims of the study are clearly defined and the study design and methods are ad equated to achieve those. |
Thank you for your approval. |
|
2. |
Page 2, third paragraph: From a cultural perspective, family obligations are the strongest barrier to engaging in regular physical activity. Regarding the different effects of sport and exercise on men and women, it has long been established that the differences come from the stereotypes on sex/gender roles [21]. Do you mean that effects of PA on men and women depend on gender roles, that engagement in PA depend on gender roles or that perception of PA depend on gender roles? Please clarify. |
Thank you for your instruction. We mean the engagement in PA depends on gender roles. We have revised the sentence accordingly. Please see page 2, lines 31-38. |
|
3. |
Please check the last paragraph on the same page: The current study had two research questions: 1) What are the factors that can be predicted who expended more activity energy in terms of MET-minutes per week among adults in rural areas? |
Thank you for the reminder. We have revised the sentence. Please see page 2, lines 49-52. |
|
4 |
Methods Page 3, Measures, 2.2.2. Influence of Physical activity. Please check. |
Thank you for the reminder. We have revised the heading and checked the content. |
|
5 |
Results, page 5, second paragraph In terms of physical activity patterns, 56.7% of the 342 participants achieved a minimum of 600 MET-min/week and were classified as ‘sufficiently active’. However, 59.4% (95% CI = 56.43–62.37) of the participants were not considered physically active. |
Thank you for the reminder. We understand how we phrased this part may have caused some confusion. Please note that ‘sufficiently active’ and ‘physically active’ are two different criteria with different definitions, and the two percentages in question are not complementary of each other (i.e., they do not add up to 100%). ‘Sufficiently active’ is defined by an IPAQ score of 600 MET-min/week. And ‘physically active’ is when people meet the ‘recommended’ level of physical activity as defined by Healthy People 2020 (please see page 5, lines 1-5). We used both definitions to categorize the participants and found that 56.7% were sufficiently active (meaning, 43.3% were not) and 40.6% were physically active (meaning, 59.4% were not). We realize that describing the first categorization as ‘meeting’ and the second categorization as ‘not meeting’ may be confusing to the reader, so we now present both as the percentage of people meeting their respective criteria. Please see page 5, lines 18-21. |
|
6 |
In the bivariate analysis, men (51.0%) were more active, with a higher prevalence ratio (PR = 2.1; p = 0.001) than women (33.2%). Higher prevalence ratio of meeting PA recommendations? |
Thank you for raising this question. Yes, we do mean the prevalence ratio of meeting PA recommendations. Generally speaking, prevalence is suitable for describing morbidity. Your question has made us realize that using this term may confuse the readers, so we have replaced all instances of ‘prevalence’ or ‘PR’ with ‘rate of odds ratios (RR)’. For the sentence in question, we revised it as follows: In the bivariate analysis, men (51.0%) were more active, with a higher rate of odds ratio (RR = 2.1; p = 0.001) than women (33.2%). Please see page 5, lines 28-33, page 6, lines 12-15, and Table 3. |
|
7. |
In other words, social support had no effect on physical activity among adults with diabetes in Indonesian rural areas. |
YES. In the current study, both the logistic regression and multiple linear regression analyses show that social support had no significant effect on physical activity. Please see Tables 3 and 4. A possible reason for the lack of impact of social support on physical activity is discussed on page 9, lines 34-48. |
|
8 |
Did you explore the model in men and in women? Some additional qualitative research on this issue would be welcome. |
We did not explore the model of physical activity for men and women separately because the sample size was not enough to support such an analysis. Thank you for suggesting that “qualitative research on this issue would be welcome”. It is a valuable comment and we have added it in the future work section. Please see page 10, lines 42-44. |
